# Seed Volume Dataset—An Ongoing Inventory of Seed Size Expressed by Volume

**Elsa Ganhão and Luís Silva Dias ***

Department of Biology, University of Évora, Ap. 94, 7000-554 Évora, Portugal; emng@uevora.pt
*   Correspondence: lsdias@uevora.pt

**Abstract:** This paper presents a dataset of seed volumes calculated from length, width, and when available, thickness, abstracted from printed literature—essentially scientific journals and books including Floras and illustrated manuals, from online inventories, and from data obtained directly by the authors or provided by colleagues. Seed volumes were determined from the linear dimensions of seeds using published equations and decision trees. Ways of characterizing species by seed volume were compared and the minimum volume of the seed was found to be preferable. The adequacy of seed volume as a surrogate for seed size was examined and validated using published data on the relationship between light requirements for seed germination and seed size expressed as mass.

**Keywords:** germination in light and dark; seed size; seed volume; soil seed banks

---

## 1. Summary

Seeds have been credited with being the main reason for the overall dominance of seed plants [1], which are recognized as the most successful group of plants in the widest range of environments [2]. It is recognized, apart from some exceptions, that plants and plant communities can only be understood if the importance of seeds in soil (seeds viewed not in a strict morphologic sense but rather encompassing fruits such as achenes, caryopses, cypselae, and others) is acknowledged, because at any moment, soil seed banks represent the potential population of plants through time [3].

The quantification, ecology, and dynamics of soil seed banks have a long history of being investigated (for example, [4,5]). With a later start, and mostly evolving in a parallel track, research on seed-functional ecology and seed-trait correlates is heavily dependent upon seed volume, which is overwhelmingly represented by the mean seed mass (for example, [6–8]).

Studies of the quantitative and qualitative composition of soil seed bank communities rarely integrate the growing knowledge on ecological correlates of seed size, probably due to the insurmountable difficulty of determining the mass of the hundreds or thousands of seeds which are easily collected when sampling soil. In our own experience, nine samples of 5 cm diameter and 20 cm depth taken from a sandy loam soil near Évora, Southern Portugal (38°32' N, 8°1' W) can yield as many as 30,000 seeds from twenty species [9].

To integrate ecological and functional correlates of seed size distributions in soil seed bank studies, seeds must be sorted by size, which for practical reasons is usually done by passing seeds and associated soil through a series of sieves of diminishing mesh sizes. However, sieves only separate seeds according to a linear dimension, rather than according to their three-dimensional volume.

Therefore, we investigated [10] size-number distributions of seeds with known length, width, and thickness after being passed through a series of sieves. We found that the geometric mean of length, width, and thickness was the most adequate one-dimensional seed size estimator, providing shape-independent measures of seed volume which were directly mirrored by the sieves mesh side. This meant that seeds with a given volume would be retained by a sieve provided that the mesh side of the sieve was equal or less than the geometric mean of the three linear dimensions of the seed. This was easily determined if seeds were assimilated to idealized ellipsoids regardless of their individual shapes.

This last assumption was investigated [11] and shown to be true, and equations to estimate seed volume were determined either when complete information on seed length, width, and thickness was available or when information on the latter dimension was absent.

Overall, the results support a subsequent ongoing program of abstracting and compiling the vast amount of published data on seed length and width, as well as on seed thickness when available, from which the dataset reported here originated.

However, the equation for estimating seed volume from seed length and seed width alone resulted from a limited number of species (eight species and a total of 400 individual records of linear dimensions), which led us to reexamine and extend it by using a tentative and preliminary unpublished version of this dataset comprising more than 6000 records of linear dimensions of seeds [12].

Considering the above, we felt that we had sufficient support to justify continuing to search and abstract published literature for linear dimensions of seeds, aiming to obtain length and width at least, as well as thickness wherever possible. In this paper we describe the present status of the resulting seed volume dataset and examine the adequacy of volume as a functional correlate using seed photoblastic responses as an example.

Our ultimate goal is to build a dataset with all the linear dimensions of seeds of all species; therefore, when seed volumes are necessary for studies on soil seedbanks, seed ecology, physiology, or evolution, just to name a few, this information could be found in the seed volume dataset (SVDS). Obviously, we acknowledge that such a dataset probably can never be fully finalized. The number of species producing seeds is finite (though we have no pretention of ever entering all of them in the dataset), however, for all practical purposes the number of possible records of seed linear dimensions is not finite. Notwithstanding this, we intend to continue updating the dataset and links to future updates are presented in this paper.

## 2. Data Description

For the most part, the data were abstracted from printed literature—essentially peer-reviewed scientific journals and books, including Floras and illustrated manuals (70% of entries). Online inventories were also used but to a lesser extent (29% of entries). The remaining data (<1% of entries) were obtained directly by the authors or by colleagues during other studies, in which case the contributions are attributed in the dataset. From the beginning, the only criteria for including a species in the dataset was the availability of trustworthy data on at least two linear dimensions of their seeds. For practical reasons only, this led to a predominance of printed sources, which declined as updates were made (online sources provided 0.8% of entries in the first version, steadily growing to 93% of new entries in the third update).

We took great care when building this dataset, however, it is possible that errors may have occurred and gone unnoticed. If errors are found or suspected, please let us know so they can be corrected in future updates of the dataset.

Data were organized in tabular form in an MS Excel® 2010 spreadsheet, which is available in the Supplementary Materials (Supplementary S1) as a non-proprietary comma-separated values (CSV) format file. The version presented here, and future updates, are also available in the same format at http://home.uevora.pt/~lsdias/SVDS_03plus.csv. Each line in the dataset presents information for one single entry of a given species and document source. Therefore, the same species may appear in multiple lines, adjacent or not.

The first column, with the heading "SVDS#", displays numeric identifiers for entries. Thus, the same species may have multiple SVDS identifiers provided that more than one entry exists for the species. Natural numbers, excluding zero, were used and for reasons related to the process of building the dataset, gaps presently exist. The maximum SVDS# is 27,548 which is larger than the number of entries (i.e., 24,208). In future updates we anticipate that such gaps will progressively disappear, but no changes will be made to the actual numbering of the entries.

The second column, with the heading "SPECIES", displays the binomial name of the species exactly as written in the document source of the entry. No attempt was made to correct possible misspellings, much less to update or correct the nomenclature, except that the specific epithet was always started with a lowercase letter. However, although no nomenclatural corrections were made in the dataset, decisions had to be made in relation to the orthographic variants in the names of several species, which are detailed in Appendix A. If possible, in the future we intend to combine the naming of species as they are now presented with nomenclatural updates and validations of names and synonymy as elaborated by *The Plant List* [13], *World Flora Online* [14] or equivalent sources.

The third column, with the heading "AUTHORITY", displays the authority for the binomial, if provided, exactly as written in the document source of the entry. No attempt was made to correct misspellings, much less to update or correct the authority or to fill in for its absence.

The fourth column, with the heading "INFRASPECIFIC", displays the infraspecific, if provided, exactly as written in the document source of the entry. No attempt was made to correct misspellings, much less to update or correct the nomenclature, except that the infraspecific epithets were always started with a lowercase letter. Authorities were included if they were available in the document source. The taxonomic level is always the sub-species unless stated otherwise. For example, *cv.* for cultivar, *f.* for form sometimes spelled in full, *var.* for variety. For some entries, usually explicitly stated in the document source, the location of flowers in the inflorescence (e.g., SVDS# 811), the size/shape of seeds (e.g., SVDS# 13820–13823) or the color of the seeds (e.g., SVDS# 13978) was also included because of the relevance that location, morphology, or polymorphism may have.

The fifth column, with the heading "SYNONYM", displays the SVDS identifier of the species for which the entry is considered a synonym in the document source used. For example, SVDS# 988, *Geranium cicutarium* is presented in the document source and used as a synonym of SVDS# 981, *Erodium cicutarium*, which is considered the valid name.

The sixth column, with the heading "FAMILY", displays the family of the species, attributed by the document source used or by us following a search through a variety of document sources, mostly Floras online or offline, when the document source of the entry failed to state the family. The rule for naming families always using the *−aceae* suffix was adopted and the correct botanical name of families is presented in the dataset regardless of the spelling in the document source used (e.g., Poaceae or Lamiaceae in the dataset instead of Gramineae or Labiatae). No attempt was made to clarify disputed families, namely Amaranthaceae and Chenopodiaceae [15,16]. For this reason, sometimes the same species appears as if it simultaneously belongs to two families, which is in fact, a taxonomic impossibility (e.g., *Chenopodium album*, SVDS# 486 and SVDS# 18067). However, in a few cases, for various reasons the family attributed by a document source was corrected, and therefore all *Cleome* were named as Cleomaceae and *Mimosa pudica* was named as Fabaceae.

The seventh column, with the heading "D/P", displays the origin of the data. In entries with "D", drawings of seeds were used and in entries with "P", photographs, or sometimes photomicrographs, were used; otherwise the seeds were measured directly by us, by colleagues or presumably by the author or authors of the document source used. For details on measurements, see the Methods section below.

The eighth column, with the heading "SOURCE#", displays numeric identifiers for sources of entries, which are listed at the end of the dataset, after the last entry. Natural numbers, excluding zero, were used, and for reasons related to the process of building the dataset, gaps exist at present. The maximum SOURCE# which is 217, is larger than the number of sources listed at the end of the

dataset, which is 212. In future updates we anticipate that such gaps will progressively disappear, but no changes will be made to the actual numbering of sources.

The ninth column, with the heading "COUNTRY/REGION", displays the country or region (for example Iberian Peninsula in entry SVDS# 2580) implicitly or explicitly attributed by the source to the entry, with synonyms excluded. No attempt was made to update the names of countries that have ceased to exist (for example Czechoslovakia in the entry SVDS# 13545).

The tenth to twelfth columns, with the headings "L (mm)", "W (mm)" and "T (mm)", display data for length, width, and thickness, respectively, with two decimal digits, and synonyms naturally excluded. Data were entered in the dataset as they were in the document sources used with a very few exceptions where we corrected what were obvious typing errors. For example, the length interval in the document source for *Verbascum litigiosum* in entries SVDS# 21507 and SVDS# 21508 is "0.7−10 mm". Clearly a comma (the decimal separator in that document source) is missing in the upper value of the range.

The thirteenth column, with the heading "EQUATION", displays numeric identifiers for the equations used to calculate seed volumes from length, width, and thickness if available. Natural numbers from one to three are used to denote the equations. For the rationale and details on measurements, see [11,12] and the Methods section below.

The fourteenth column, with the heading "VOLUME (mm$^3$)", displays the volume for each entry, with synonyms naturally excluded.

The fifteenth and last column, with the heading "UPDATE#", displays numeric identifiers for updates of the dataset. Natural numbers were used and zero was included to denote the creation of the dataset, when an abridged version of it was first made available online in PDF format (May 2013, with 10,663 entries). The first update, UPDATE# 1, was completed in February 2015 (17,781 entries); the second, UPDATE# 2, in August 2015 (18,852 entries); and the third and most recent, UPDATE #3, was completed in September 2017 (24,208 entries) as a PDF file and replaced in December 2018 by a CSV file.

## 3. Methods

### 3.1. Measurement of Seeds and Determination of Seed Volume

When more than two seeds were present in a drawing or photograph, including photomicrographs, the smallest and the largest seeds were visually identified and measured with digital calipers to the nearest 0.01 mm. Direct measurements in seeds were also performed with digital calipers or with stereomicroscopes equipped with calibrated eyepiece micrometers (for details see [11,17]). Data on length and width, and less frequently also of thickness, are usually presented in document sources as intervals, for example $6 − 8 \times 4 − 5 \times 4 − 5$ mm as in entries SVDS# 1 and SVDS# 2 (*Abus precatorius*). We opted to consider that the smallest seed would have dimensions at lower end of all intervals, thus $6 \times 4 \times 4$ mm, and the same for the largest, thus $8 \times 5 \times 5$ mm. Whenever a documental source simultaneously provided drawings or photographs, including photomicrographs, and values of measurements only the latter were used in the dataset.

At this stage of development of the dataset there is no mention of seeds' appendages. In the future we intend to fill this gap either by registering their presence or by providing data on their dimensions separately from data on seeds' length, width, and thickness.

Seed length was defined as the largest linear dimension of a seed, and seed width as the largest linear dimension of the same seed orthogonal to the length-axis at its mid-point. Seed thickness was defined as the largest linear dimension of the same seed orthogonal to both the length- and the width-axes at their mid-points.

A binary decision tree was used to select the equation to be used to determine seed volume for each entry. Details and the rationale for the equations and for the binary decision tree can be found elsewhere [11,12]. Expressing the decision tree as a dichotomous key:

1.　　Values available for *L*, *W*, and *T*　　　　　　Equation (1)
　　　Values available only for *L* and *W*　　　　　　**2**
2.　　$\pi LW^2/6 \leq 4.719$ mm$^3$ and $W \leq 1.695$ mm　　Equation (2)
　　　$\pi LW^2/6 \leq 4.719$ mm$^3$ and $W > 1.695$ mm　　Equation (3)
　　　$\pi LW^2/6 > 4.719$ mm$^3$　　　　　　　　　　Equation (3)

And the equations are:

$$VOL = \pi LWT/6 \tag{1}$$

$$VOL = (0.56822\ VOL_{LW}) + (0.05156\ VOL_{LW}{}^2) \tag{2}$$

$$VOL = (0.46753\ VOL_{LW}) + (2.65493 \times 10^{-5}\ VOL_{LW}{}^2) \tag{3}$$

with

$$VOL_{LW} = \pi LW^2/6 \tag{4}$$

where *VOL* represents the volume of seeds in mm$^3$; *L*, *W*, and *T* are the length, width, and thickness of seeds in mm, as defined above; $VOL_{LW}$ is an intermediate estimate of *VOL* when only length and width are known and $W = T$ has to be assumed.

### 3.2. Characterization of the Dataset

This third update of the dataset has a total of 24,208 entries, comprising 15,376 entries (64%) with valid names according to their documental sources, and 8832 entries (36%) of synonyms. Hereafter, only entries with names assumed as valid by their documental sources will be considered.

Measurements of drawings or photographs of seeds, including photomicrographs, were provided for 51% of entries. Overall, in 1280 entries (8%) volumes derived from the three linear dimensions described above, length, width and thickness, were calculated using Equation (1). In the remaining 14,096 entries (92%) volumes were derived from length and width alone, and Equations (2)−(4) were used.

A total of 220 families are listed in the dataset. Only four families had 5% or more of the entries, namely Fabaceae with 2397 entries (16%), Poaceae with 2003 entries (13%), Asteraceae with 1526 entries (10%), and Cyperaceae with 787 entries (5%). Conversely, 92 families (42%) had only 5 entries or fewer and 19 families (9%) had only one entry. Families are grouped under 55 different orders (identified using *World Flora Online* [14], not displayed in the dataset). Five orders had 5% or more of the entries, namely Poales with 2927 entries (19%), Fabales with 2399 entries (16%), Asterales with 1628 entries (11%), Lamiales with 1504 entries (10%), and Caryophyllales with 978 entries (6%). Conversely, 11 orders (20%) had only 5 entries or fewer and 2 orders (4%) had only one entry.

A total of 150 countries or regions are represented in the dataset, after combining entries for several of them. Most notably, Portugal, Spain and Iberian Peninsula were combined in a single group (when explicitly identified, the Azores and Canary Islands were left separate). Conversely, entries for the Hawaiian Islands and Puerto Rico on one hand and the continental US with Alaska included on the other, were not combined. Putting aside the 457 entries (3%) from unknown or undetermined locations, only four regions had 5% or more of the entries; namely the Iberian Peninsula with 4865 entries (33%), the US with 4514 entries (30%), the Hawaiian Islands with 1536 entries (10%), and Brazil with 811 entries (5%). In the long tail of the frequency distribution of entries, 80 countries or regions (53%) had only 5 entries or fewer and 13 countries or regions (9%) had only one entry.

The dataset comprises 6544 species. Four families individually contained 5% or more of the total species, with only minor differences in relation to the distribution of entries. In terms of the number of species, rather than entries, Poaceae ranked first with 958 species (15%), followed by Fabaceae with 877 species (13%), Asteraceae with 546 species (8%) and Cyperaceae with 387 species (6%). Conversely, 100 families (46%) were represented by 3 species or fewer and 59 families (27%) by a single species.

The distribution of the number of entries per species was strongly asymmetric and skewed to the right (coefficient of skewness $g_1 = 3.985$, $P \approx 0$; for details on statistics see Appendix B) and ranged

from 1 entry in 1611 species (25%) to 24 entries in *Trifolium subterraneum*, in the latter species almost all entries from Australia and the Iberian Peninsula. The mean number of entries per species (± standard error) was 2.3 ± 0.02, and the median and mode were both 2—the mode with 56% of entries. Altogether, more than two-thirds of species (68%) had 2, 3, or 4 entries each. In addition to the 1611 species that had only one entry, the volumes of the two entries of six species (*Bombycilaena erecta*, *Chamaemelum fuscatum*, *Gymnostyles stolonifera*, *Hydrocotyle bonariensis*, *Pulmonaria longifolia*, and *Rumex thyrsoides*) were exactly the same.

The orders of magnitude (see Appendix B for details) of seed volumes ranged from −4 (*Hydrilla verticillata*, *Striga lutea*, and *Tofieldia pusilla*) to 6 (*Terminalia kaernbachii*). However, the within species variation of the order of magnitude was much lower. Considering only species with two or more different entries, in 3898 (79%) the minimum and maximum volume were of the same order of magnitude and in 995 species (20%) the minimum and maximum volume varied only by one order of magnitude.

### 3.3. How to Express Seed Volume?

For better or worse, whenever a species in the dataset has only one entry (or the same volume if more than one entry) there is no question as to how to express its seed volume because, given the available information and for all practical purposes, the solitary value is necessarily the best possible estimate of the seed volume of the species

In the remaining species, decisions on how to express the seed volume must be taken. A good criterion is that seed volume should be maximally independent from environmental influences in seed formation and in volume build up, so that it can be maximally representative of the species regardless of the origin of seeds or of the exogenous conditions under which the seed developed.

Seed size is a highly hereditary characteristic [18,19] and the very low plasticity of seed size could be, not exclusively but in large measure, a consequence of the short time available for their formation in comparison with the more extended periods available for the formation of more plastic organs [20]. Conversely the number of seeds per plant has a much higher plasticity being, together with ovary length, one of the few characteristics of reproductive systems in plants with real plasticity [21].

Nevertheless, the amplitude of seed size within species is likely to increase with the abundance of resources, with the minimum size being less affected by the availability of resources [22,23]. Therefore, it can be hypothesized that the amplitude of seed volume within species depends more on the maximum volume than on the minimum.

This hypothesis was tested with the data obtained from the dataset (Figure 1) by regressing the amplitude of volumes on the minimum volume, and on the maximum volume, with all variables transformed logarithmically (base 10).

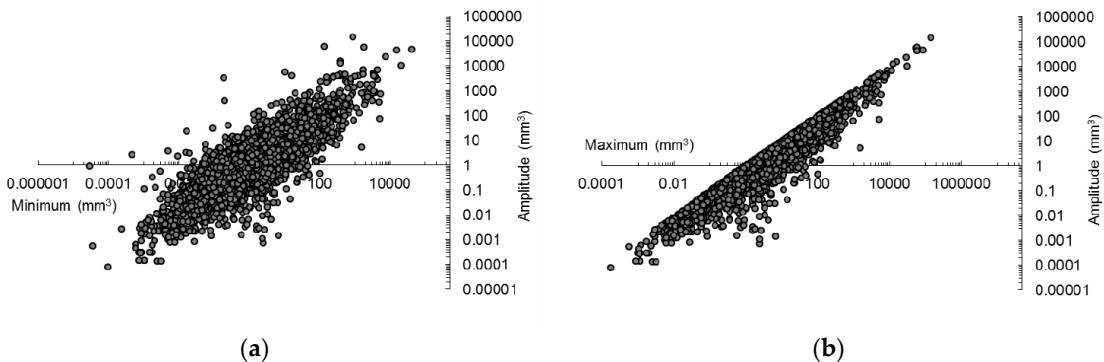

(**a**)　　　　　　　　　　　　　　　　　　(**b**)

**Figure 1.** Biplots of: (**a**) minimum seed volume and amplitude of seed volumes, (**b**) maximum seed volume and amplitude of seed volumes of the 4927 species in the dataset with two or more different values of seed volume.

Linear regressions were always significant ($P \approx 0$) but the amount of the variation of amplitudes of seed volume statistically explained by the minimum volume, expressed by the coefficient of determination adjusted for degrees of freedom, was much lower ($R^2_{adj} = 0.688$) than the amount of variation explained by the maximum volume ($R^2_{adj} = 0.893$). Therefore, the minimum value is clearly the best way to express seed volume.

*3.4. Does Seed Volume Qualify as an Ecological Correlate? Testing its Adequacy with Published Data on Light Requirement for Seed Germination*

Milberg et al. [24] investigated the germination under light and darkness of 54 species suspected or known to accumulate persistent seed banks in soil. Seeds were cold-stratified to simulate winter and results were assumed to simulate a spring situation in a ruderal plant community. Seed size was expressed as average seed mass and ranged from 0.032 mg (*Conyza canadensis*) to 22.2 mg (*Avena fatua*). The light requirement of seeds to germinate was expressed as relative light germination (*RLG*):

$$RLG = G_L/(G_D + G_L) \tag{5}$$

where $G_L$ and $G_D$ are percentages of germination in light and dark respectively. *RLG* can vary from 0 ($G_L = 0\%$, $G_D > 0\%$; germination in darkness only) to 1 ($G_L > 0\%$, $G_D = 0\%$; germination in light only). The authors regressed *RLG* on average seed mass transformed logarithmically (base 10) and found that the straight line

$$RLG = 0.7402 - 0.1577 \log_{10} M \tag{6}$$

where *M* is the average seed mass, was highly significant ($P < 10^{-4}$). The coefficient of determination was 0.2774 which means that the variation of $\log_{10} M$ statistically explained approximately 28% of the variation of *RLG*.

To evaluate the adequacy of seed volume we analyzed *RLG* using the minimum volume in the dataset. However not all species used by Milberg et al. are present in the dataset, and *RLG*-values of *Consolida regalis*, *Galeopsis bifida*, *Chenopodium polyspermum*, *Chamomilla suaveolens*, *Chenopodium suecicum*, *Filaginella uliginosa* and *Rumex longifolius* could not be used (*RLG*-values between 0.5 and 1). For *Chamomilla recutita* we used the volume of *Matricaria chamomilla* after checking the synonymy in [13,14].

To set a common base for comparison, first we investigated the relationship between *RLG* and the minimum seed volume for which data existed in the dataset, and recalculated Equation (6) using only the 47 species for which we had volumes and obtained the straight line (Figure 2a):

$$RLG = 0.7338 - 0.1537\log_{10} M \tag{7}$$

which was also highly significant ($P = 0.0002$). The coefficient of determination was 0.2634 (0.2470 when adjusted for degrees of freedom) which means that the variation of $\log_{10} M$ statistically explained approximately 26% of the variation of *RLG*, almost the same amount that was obtained for the whole set of 54 species.

The relationship between *RLG* and the minimum volume was investigated by stepwise regression (details in Appendix B) and the fitted equation was:

$$RLG = 0.0706 - 0.0793 \log_{10} V_{min} \tag{8}$$

where $V_{min}$ is the minimum seed volume in mm$^3$ (Figure 2b). The equation was significant ($P = 0.0201$), all coefficients with $P \leq 0.0201$. The coefficient of determination was 0.1142 (0.0945 when adjusted for degrees of freedom) which means that the variation of $\log_{10} V_{min}$ statistically explained approximately 11% of the variation of *RLG*, less than the amount explained when the mass of the seeds actually tested was used. However, contrary to mass-values, volumes now came from seeds that had little or no relation to those used to determine *RLG*. Nevertheless, the main conclusions are unchanged.

In short, minimum seed volume is promising for use as an alternative to seed mass in studies of ecological correlates of seed size. Nevertheless, further investigations on the adequacy of seed volume are highly advisable but are clearly beyond the scope of this paper.

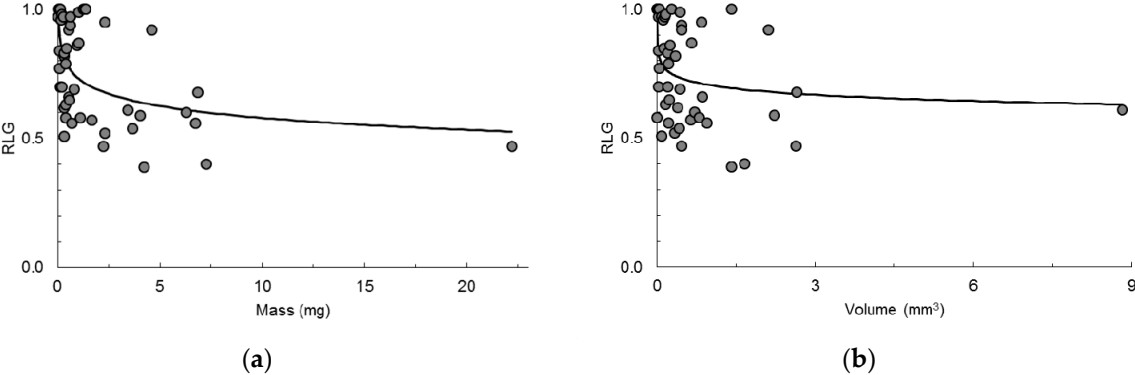

(**a**)　　　　　　　　　　　　　　　　　　　　　　　(**b**)

**Figure 2.** Observed and estimated relationship between relative light germination (*RLG*) and (**a**) seed mass, (**b**) minimum seed volume. *RLG* and seed mass from [24], seed volume from the dataset reported here.

**Supplementary Materials:** The following are available online at http://www.mdpi.com/2306-5729/4/2/61/s1, Supplementary S1: Seed Volume Dataset update 03 September 2017.

**Author Contributions:** Conceptualization, data curation, formal analysis, methodology, resources, writing—original draft, L.S.D.; Investigation, E.G.; Writing—review and editing, E.G. and L.S.D.

**Funding:** This research received no external funding.

**Acknowledgments:** We thank the Herbarium of University of Évora for administrative and technical support, and one reviewer for comments and suggestions on an earlier version of this paper.

**Conflicts of Interest:** The authors declare no conflict of interest.

## Appendix A

A number of binomials raised doubts on whether they were orthographic variants, typing errors, or different species and were checked mostly using [13,14] and also going through their document sources. Therefore, and despite no corrections being made in the dataset, the following (displayed in alphabetic order) were taken here as being the same species: *Ambrosia artemisiaefolia/Ambrosia artemisiifolia; Amphicarpa bracteata/Amphicarpaea bracteata; Arachis hypogaea/Arachis hypogea; Baccharis cordifolia/Baccharis coridifolia; Bauhinia monandra/Bauhinia monarda; Borago officinalis/Borrago officinalis; Carex nebrascensis/Carex nebraskensis; Centrosema virginiana/Centrosema virginianum; Chenopodium ambrosioides/Chenopodium ambrosoides; Cistus ladanifer/Cistus ladaniferus; Cladrastis lutea/Cladratis lutea; Conringa orientalis/Conringia orientalis; Cuscuta europaea/Cuscuta europea; Ecbalium elaterium/Ecballium elaterium; Echinochloa colona/Echinochloa colonum; Echinochloa crusgalli/Echinochloa crus-galli; Erechtites valerianaefolia/Erechtites valerianifolia; Eriochloa contracta/Eriochloa contractra; Eschscholtzia californica/Eschscholzia californica; Euphorbia lathyris/Euphorbia lathyrus; Gymnocladus dioica/Gymnocladus dioicus; Hoffmannseggia glauca/Hoffmanseggia glauca; Hypochaeris radicata/Hypochoeris radicata; Lotus corniculata/Lotus corniculatus; Lupinus cosentini/Lupinus cosentinii; Medicago lupilina/Medicago lupulina; Melilotus alba/Melilotus albus; Melilotus indica/Melilotus indicus; Myosotis arvense/Myosotis arvensis; Nicandra physalodes/Nicandra physaloides; Ornithopus compressus/Ornithopus compresus; Pachyrhizus erosus/Pachyrrhizus erosus; Physalis heterophyla/Physalis heterophylla; Schinus terebinthifolius/Schinus terenbinthifolius; Scorpiurus sulcata/Scorpiurus sulcatus; Scorpiurus vermiculata/Scorpiurus vermiculatus; Setaria faberi/Setaria faberii; Silene cucubalis/Silene cucubalus; Sinapis arvensis/Sinapsis arvensis; Sonchus oleraceous/Sonchus oleraceus; Sphenoclea zeylandica/Sphenoclea*

*zeylanica*; *Synedrella nodiflora/Synedrella nodifora*; *Trigonella foenum-graceum/Trigonella foenum-graecum*; *Urtica dioca/Urtica dioica*; *Veronica hederaefolia/Veronica hederifolia*.

On the contrary, *Acacia tortuosa* and *Acacia toruosa* were considered different species. The same applies to *Erechtites hieracifolia* and *Erechtites hieraciifolius*, and to *Trigonella polycerata* and *Trigonella polyceratia*.

## Appendix B

Orders of magnitude of seed volumes in Section 3.2 were determined as the numeric value resulting from the truncation of the decimal logarithm of the volume.

Least squares linear regressions without replication in Section 3.3 were done with an experiment-wise error rate for coefficients of 0.05 calculated by the Dunn-Šidák method [25].

The relationship between *RLG* and seed mass or seed volume was investigated in Section 3.4 by stepwise least squares regression without replication and an experiment-wise error rate for coefficients of 0.05 calculated by the Dunn-Šidák method [25]. Polynomials were used as candidate models and included up to the third power of the explanatory variables either untransformed or logarithmically transformed (base 10). For each explanatory variable (seed mass or seed volume) the criterion for model selection was the coefficient of determination adjusted for degrees of freedom.

Determinations of skewness and regression analyses were done with Statgraphics 4.2 (STSC, Inc., Rockville, MD, USA), and all other statistics with MS Excel® 2010 (Microsoft Corporation, USA).

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
