# Peer review of "Seed Volume Dataset—An Ongoing Inventory of Seed Size Expressed by Volume"

_data_

Round 1
Reviewer 1 Report
Seed Volume Dataset − an Ongoing Inventory of Seeds’ Size Expressed by Volume
The article presents a dataset of seeds including information relative to name, origin, and dimensions of about 24000 seed samples taken from different sources, mainly scientific literature, articles and books. It is part of the continued research of the authors on seed banks and is preceded by previous work in this field.
Data from the seeds does not include important aspects such as seed shape and appendices or seed ornaments. These are aspects that the authors need to consider to include in future versions of the database. Dimensions include length and width in all samples; thickness in some of them. With these data the authors have calculated by a theoretical approximation the volume of the seeds, a data that is also included in the database.
The subject is interesting and the technical approach adequate. This review is divided in two parts. The first part contains important corrections on major aspects (conceptual) that the authors need to take into account to prepare the corrected version. Part two contains minor corrections easy to do, that must be done also.
Part one (main conceptual questions):
The database contains 24,208 entries. Those with valid names are 15,376 entries (64 %), and these are the only ones considered. A major point concerns the origin of data. The abstract makes emphasis on soil seed banks, but the entries in the database come from “printed literature, essentially peer-reviewed scientific journals and books, including Floras and illustrated manuals (77% of entries)” as stated in the first paragraph of section on Data Description. The authors need to explain clearly what is the relation, if any, of entries in the database with soil seed banks or expressed in another way: what has been the criteria to select these species for their analysis. There are many seed image databases that could be useful for their purpose, but the fact that the seed data are taken mainly from articles (and books) makes to think that there may be a criterium for selection. Is there such criterium? If so, please explain it. This important information about the database is missing at the moment.
The authors detail the number of families and species to which entries belong. Entries correspond to a total of 220 families and 6,544 species. Please could you check to how many orders do these families belong? please include the data in the article.
Overall there is a lack of consideration with seed morphology and seed shape. I would advise to include an entry in the database with specifications related to seed shape (cardioid, ellipse, ovoid, all these are geometric figures, well defined, but seeds may present other shapes: irregular, fusiform, filiform…). This is not required in the present review but may be taken as an advice to improve the database.
Part two (details, also important):
There are some questions that need to be corrected. Please explain and/or correct the following sentences:
1. In the Abstract:
The abstract need to reflect very well what is each magnitude: length, width, thickness are lineal magnitudes, expressed in lineal units (mm, micrometers); volumen is a tri-dimensional magnitude measured in cubic milimeters, or cubic micrometers.
These two sentences need to be corrected:
This paper presents a dataset of seeds’ volumes calculated length, width and, when available, thickness abstracted from printed literature, essentially scientific journals and books
Seeds’ volumes were determined from the linear of seeds using published equations and decision trees.
2. In the summary:
Please make clear:
Preying in our own experience, nine soil samples 5 cm and 20 cm depth can yield as much as near 30,000 seeds from twenty species
Does this refer to nine soils form the same location or nine soils from a different location each. In any case soils from which location (s)?
Why do you thing the geometric mean of seeds’ length, width, and thickness is better estimator than the arithmetic mean of these values?
3. In Data Description:
It is stated:
the size of seeds (e.g. SVDS# 13820) or the color of 108 seeds (e.g. SVDS# 13978) because of the relevance that location, morphology, or heteromorphy may have.
Size has no much relevance in morphology, because shape is independent of size.
Similarly color has no relevance in heteromorphy.
3.1. Measurement of seeds and seeds’ volume determination
largest seeds were visually identified and measured without appendices,
As already mentioned it may be important to annotate that these appendices forms will be taken in consideration in future versions of the database.
In section 3.3. How to express seed volumes?
"...and the very low plasticity of seeds size would be in large measure a consequence of the short time available for their formation in comparison with the more extended periods available for the formation of other, much more plastic organs like leaves [20]."
Don’t agree. Leaves, independently of time for their formation have much more plasticity than seeds. Seeds have much more size requirements than leaves do. Growth of seeds is spatially restricted, growth of leaves is not.
Please check data points to the left in figure 1a. These are very strange because with 0.000001 mm3 vol give amplitudes in the range of 1 to 10 mm3. Correct if it is incorrect or explain this if it is OK.
Reviewer 2 Report
In my opinion, the manuscript provides detailed and valuable information that raises no doubts regarding its quality or reliability, and the presentation of data conforms to the relevant methodological and formal standards.
I have no major content-related or methodological concerns regarding the submitted manuscript. All important aspects of the study that is to be published as Data Description, including data description, data quality, data access, archiving and meta data, are correctly presented and do not require revisions or modifications. The information provided is sufficient and does not have to be supplemented. In my opinion, the manuscript is acceptable for publication in the Data Journal in its present form.
Round 2
Reviewer 1 Report
The article has been improved and deserves publication.